# On-field Gross Morphology Evaluation of Dromedary Camel (*Camelus dromedarius*) Fetal Membranes

**DOI:** 10.3390/ani14111553

**Published:** 2024-05-24

**Authors:** Davide Monaco, Carolina Castagnetti, Aliai Lanci, Taher Kamal Osman, Giovanni Michele Lacalandra, Jasmine Fusi

**Affiliations:** 1Department of Veterinary Medicine, University of Bari Aldo Moro, Str. Prov per Casamassima, 70010 Bari, Italy; davide.monaco@uniba.it (D.M.); giovannimichele.lacalandra@uniba.it (G.M.L.); 2Department of Veterinary Medical Sciences, Università degli Studi di Bologna, Via Tolara di Sopra 50, 40064 Bologna, Italy; carolina.castagnetti@unibo.it; 3Department of Advanced Biotechnology and Research, Salam Veterinary Group, Buraydah 51911, Saudi Arabia; taher.kamal@svg.sa; 4Department of Veterinary Medicine and Animal Sciences, Università degli Studi di Milano, 26900 Lodi, Italy; jasmine.fusi@unimi.it

**Keywords:** dromedary camel, placenta evaluation, chorion, allantois, umbilical cord, linear measurements, weight, volume

## Abstract

**Simple Summary:**

Simple Summary: The post-partum gross examination of the fetal membranes is an efficient tool for detecting the abnormalities that might influence the survival, health, and future development of a newborn. The routine on-field gross evaluation of the placenta performed by veterinarians could represent a valuable tool for the management of dromedary camel (*Camelus dromedaries*) newborns. A description of the dromedary camel placenta was therefore performed in this study. Female dromedary camel parturitions were attended and placentas were collected, cleaned of sand, and observed. The chorionic surface appeared velvety and non-uniform in color, varying from pink to pale or bright red and often with dark red areas; the latter were located mainly around the uterine body portion or in the area outside the insertion of the umbilical cord. The pregnant horn portion was always bigger in size than that of the non-pregnant portion; the uterine body portion showed variability in shape and size. The allantoic surface showed variable appearance, from pearly white, to pink or red-bluish, depending on the background color of the chorionic surface, and was characterized by the presence of several blood vessels.

**Abstract:**

The dromedary camel (*Camelus dromedarius*) fetal membranes, commonly referred to as “the placenta”, are epitheliochorial, diffuse, and microcotyledonary, similarly to the mare’s placenta. The evaluation of the placenta is an essential component of the neonatal evaluation in the equine species. However, post-partum or post-abortion placental assessment in dromedary camels is unfortunately too frequently neglected and, to the best of the authors’ knowledge, the dromedary camel species lacks a comprehensive description of the normal placenta’s gross morphology. In order to facilitate its on-field evaluation, the current study describes the macroscopic features of the placenta of the dromedary camel after full-term pregnancy and spontaneous parturition.

## 1. Introduction

The placenta has been defined as the apposition, or fusion, of the fetal membranes to the uterine mucosa; this process can, in some cases, suffer from alterations both from local or systemic diseases. However, little is known about these pathological aspects in camelids, especially dromedary camels. The camel placenta is epitheliochorial, with no invasion but an apposition of the trophoblast after implantation [1]. Interdigitations from the two structures starts from day 25 after ovulation [2]. It is common, however, to refer to the fetal membranes as “the placenta” [3].

During intra-uterine life, the placenta provides the interface necessary for metabolic exchange (i.e., oxygen, nutrients, and waste material) between the dam and the fetus, for protecting the fetus from external and internal insults and, in some species, e.g., pigs, cats, and horses, for the production and metabolism of various hormones [3,4,5]. Previous studies have reported that the proper anatomical and functional development of the feto-maternal interface is essential for the successful establishment and the normal course of pregnancy: pathologic placentas are associated with several pregnancy complications (abortion, stillbirth, premature births, fetal hypoxia) [6].

Among domesticated animals, no other animal can provide such a variety of products (milk, meat, skin), leisure activities (racing, festival, tourism), and agricultural work, as the dromedary camel—*Camelus dromedarius* (DC). In pastoral areas, the camel is able to graze the greatest variety of plants, in comparison with other herbivorous ruminants, including halophyte grasses, bushes, and trees, leading to lower pressure on the floristic biodiversity of the arid lands [7]. Nowadays, DC breeding is moving from the traditional extensive breeding system (low-input and scarce veterinary care to more of a farming system for meat and milk production [8]). However, despite a marked improvement in some DC reproductive technologies, some basic aspects have been poorly considered. For instance, the evaluation of the placenta is considered a pivotal part of the postpartum examination of dams and newborns in many species, and in controlled conditions such as a “camel farm” its retrieval and evaluation could make it a valuable tool for the postpartum care of the dams and the calves.

Other than identifying possible causes of abortion, the gross placenta examination could allow the prompt recognition and management of the newborns in need of specific care: in mares, the post-partum gross examination of the placenta is an efficient tool for detecting abnormalities that might influence the survival, health, and future development of the newborn [9,10,11,12].

In the Camelidae family, the placenta is epitheliochorial, diffuse, and microcotyledonary, therefore similar with the equine placenta [13,14]. However, to the authors’ knowledge, differently from horses and other equids, no information to date is available about the gross morphology features of the normal placenta and its efficiency in DCs.

Although some literature about the histology and ultrastructure of the camelid placenta exists [2], most of the time clinicians must cope with on-field conditions, with the need of criteria immediately recognizable to provide the best assistance to dromedary camel newborns. The routine on-field gross evaluation of the placenta, performable by veterinarians even under intensive breeding systems, could represent a valuable tool for the reproductive management of DC dams and their newborns. The objective of the present study was, therefore, to describe the normal gross morphological features of the dromedary camel placenta, collected after spontaneous parturition, and evaluated on-field.

## 2. Materials and Methods

### 2.1. Animals and Inclusion Criteria

The study was performed in Al-Qassim region (Saudi Arabia), during March–April 2022, and involved pregnant camel recipients (8–15 years of age) enrolled in an embryo transfer program. All animals were previously screened and declared free from brucellosis and were kept in a free barn with shaded areas. They received alfalfa two times a day, had free access to water, and were allowed to graze on desert pastures for at least three hours per day. The following conditions were required to select normal placentas: (a) eutocic, spontaneous parturition at term of pregnancy; (b) expulsion of the complete placenta within three hours after the calf birth; (c) delivery of a calf that was reported to be healthy and fit after clinical examinations carried out daily, during its first week of life. The gestation length (GL) was calculated after parturition considering the day of the donor mating as the beginning of the pregnancy, and only placentas from pregnancies lasting from 350 to 400 days were included in the study.

### 2.2. Data Collection

Parturitions were attended, but without intervention. Soon after the delivery of the calf, and before the first suckling, the calf birth weight (BW) was recorded to the nearest 100 g, with a portable hanging scale (Sunrise Technology**^®^**, Ahmedabad, India). The time elapsing between the delivery of the calf and the expulsion of the placenta was recorded, and each placenta was put in a plastic bag and transported at room temperature to the Salam Veterinary Hospital.

#### Placental Measurements

The fetal membranes were evaluated within 4 h from expulsion; before the evaluation, they were gently and rapidly rinsed with cold running water to remove sand debris, then they were laid down flatly on the floor, with the chorionic surface above, forming a “Y” shape. After visual inspection, the description of the different placenta components (chorion, allantois, amnion, and umbilical cord) were performed; therefore, the following linear measurements were collected with a measuring tape (cm graduated), according to the scheme proposed by Whitwell and Jeffcott [15] for the mare, and adapted to the DC: length of the pregnant horn (LPH); cranial diameter of the pregnant horn (CDPH); middle diameter of the pregnant horn (MDPH); caudal diameter of the pregnant horn (CaDPH); length of the non-pregnant horn (LNPH); cranial diameter of the non-pregnant horn (CDNPH); middle diameter of the non-pregnant horn (MDNPH); caudal diameter of the non-pregnant horn (CaDNPH); length of the uterine body (LUB). After the evaluation of the chorionic side, the FMs were turned inside out, the allantoic side was inspected, and the umbilical cord was freed from the amniotic sac and measured in length, from its horn attachment to its free end (LUC) (Figure 1). After the linear measurements and the evaluation of the chorionic and allantoic sides, the placentas were photographed from a minimum height of 3 m, by using a digital camera.

### 2.3. Statistical Analysis

Data were submitted to the Shapiro–Wilk test for normality assessment. Afterwards, the Pearson correlation test was used to assess possible correlations among the following parameters: calf BW, TPW, PTV, VPH, VNPH, VUB, and VUC. The *t*-test was used to assess possible differences between males and females in the above-reported parameters. Descriptive data were expressed as mean ± SD, median, and min–max.

## 3. Results

Eleven parturitions fulfilled the selection criteria and 11 FMs were therefore evaluated for the study. In some cases, the fetal membranes were partly retained because of the tearing of the non-gravid (right) uterine horn portion at its middle level or at the level of its caudal diameter (Appendix A); the retained right horn part was, however, expelled within 4 h: those FMs were excluded from the study. After the delivery, the fetal membranes were expelled with the allantoic surface of the pregnant horn portion (always the left) filled with fluid and facing outside, and with the right uterine portion facing the chorionic surface out.

Descriptive data about pregnancy length, calf expulsion time, placenta expulsion time, male calves BW, and female calves BW are reported in Table 1.

### 3.1. Gross Description of Different Placental Components

After cleaning and positioning, the chorionic surface of the FMs appeared velvety and non-uniform in color, varying from pink to pale or bright red and often with dark red areas; the latter were located mainly around the uterine body portion or in the area outside the insertion of the umbilical cord (Figure 2(2A–6A)).

The chorionic velvety surface appeared almost uniform; macroscopic avillous areas (i.e., cervical star or in correspondence with the folding of the chorion around the allantoic vessels) were not noticed. The pregnant horn portion was always bigger in size than that of the non-pregnant portion; the uterine body portion showed variability in shape and size. No macroscopic edema was observed on the chorionic or allantoic side. The allantoic surface of fetal membranes showed variable appearance, from pearly white to pink or red-bluish, depending on the background color of the chorionic surface, and was characterized by the presence of several blood vessels (Figure 2(2B–6B)). The allantois occupied the non-pregnant horn portion, part of the uterine body, and part of the pregnant horn portions. The amniotic sac was composed of a part adhering to the allantochorion and of a free part. The attachment of the amnion to the allantochorion was found to be on the ventral part of the pregnant horn portion (Figure 2(3B–6B)) (i.e., corresponding to the dorsal part of the uterine wall, in situ) but it was also sometimes observed on the upper surface of the spread placenta (Figure 2(2B), Appendix A); its extension was limited to the pregnant horn portion and it was never detected beyond the limit of the corresponding side of the uterine body portion.

The amnion color varied from white to yellow to yellow-greenish; this latter color was noticed in concomitance with the presence of meconium (samples excluded from the study; Appendix A). The umbilical cord was included in the amniotic sac, and it was composed of two arteries, two veins, and the urachus. Amniotic plaques (dispersed focal, slightly raised, 1–3 mm small pinpoint plaques) around the umbilical cord attachment at the fetal side of the amnion were noticed in two out of eleven samples (Figure 3).

Camelmanes (hippomanes) (measuring about 4 × 2 cm) (Figure 4) were found in 50% of the observed specimens.

In all the observed placentas, the umbilical cord attachment was always located at the middle of the pregnant horn portion in correspondence to its medial side; in one case, the attachment was found to be on the lateral side of the pregnant horn portion (Appendix A). This sample, however, was excluded from the study. The umbilical vessels were located on the medial surface of the pregnant uterine horn portion with one branch running toward its apex and the other one running toward the uterine bifurcation and reaching the tip of the non-pregnant horn portion; their diameter decreased as the distance from the insertion point of the umbilical cord increased (Figure 5).

In one fetal membrane, an avillous area on the chorionic surface in the pregnant horn portion was found (Figure 6), attributed to a fetal foot placenta area of degeneration (PAD) resulting from chronic pressure of the fetal foot against the placenta, as similar with the observations of Schlafer [17,18] in the equine species.

### 3.2. Placental Measurements

The mean weight (g) ± SD of the selected placentas TPW was 6364.1 ± 1488.55 (median: 6190.5; min 4547.0–max 9638.0). Table 2 provides descriptive information about the linear measurements (cm), which are expressed as mean ± SD (min–max) values. Table 3 reports the total volume as well as the volumes of the uterine body and uterine horns.

The statistical analysis showed significant correlations between TPW and TV (r = 0.87; *p* < 0.01), between PTW and VPH (r = 0.93; *p* < 0.001), between TPW and VNPH (r = 0.66; *p* < 0.05), between TPW and VUB (r = 0.69; *p* < 0.05), between TV and VPH (r = 0.78; *p* < 0.01), and between TV and VNPH (r = 0.72; *p* < 0.05). Moreover, significant correlations were also found between CaDPH and MDPH (r = 0.62; *p* < 0.05), between CDNPH and MDNPH (r = 0.64; *p* < 0.05), between the CaDNPH and MDNPH (r = 0.87, *p* < 0.001), and between LUC and LPH (r = −0.71, *p* < 0.05).

Male and female calf weights were not statistically different. However, when differences between placental components and calf gender were considered, the mean TPW (7687.8 ± 1428.99 vs. 5481.7 ± 653.55 cm^3^, *p* < 0.05) and VPH (4802 ± 898.62 vs. 3025 ± 343.15 cm^3^, *p* < 0.01) was significantly higher in males than females.

Placental efficiency, calculated as TPW/calf BW, was 15%, and the BW/TPW was 6.7.

## 4. Discussion

Due to the seasonality, the late age at first mating (4 years), and the gestation length (13 months), a DC female could produce a single calf every two or more years [19]. Given the multiple attitudes of DCs and their role as a food source in marginal areas, the survival of DC calves is acquiring greater importance over time: the calves’ mortality significantly threatens the sustainability of the camel farming systems. A simple procedure for an initial on-field evaluation of the placenta is essential to recognize possible macroscopic modifications that could, in turn, be responsible for or influence calves’ viability and health. This allows to provide prompt and specific care, as reported in other species. To the authors’ knowledge, this is the first proposal of a standard method for placenta evaluation in the dromedary camel species. Other than this, a description of gross anatomical features was provided. This first descriptive report can also be pivotal for future studies, in which ecotype-specific measurements could be identified as well as other factors that might affect the development of the placenta. In addition, the identification and classification of abnormalities can be put in relation with calf health and survival to distinguish incidental from pathological findings.

The spontaneous deliveries gave birth to alive and healthy calves, therefore supporting the observations of normal gestational lengths and of the calf body weights, even though a slightly shorter GL was observed as compared to a previous study [20]. The neonatal BW was within the ranges observed by Athien et al. [21] and Al-Mutairi [22] in Saudi Arabia and higher than the average BW recorded in the Saudi ecotype calves (34.5 kg) [20]. The Saudi Arabian breeder’s selection for big-size animals might explain the ecotype differences, which in turn could influence the newborn BW. Moreover, because all the calves were born from embryo transfer programs, other than the origin of the embryo, the recipient size might have also affected the newborn BW, as previously reported in the horse [23]. The difficulties in recording the donor and the recipient body linear measurements in this preliminary study have limited the obtainment of such information; however, this aspect is worthy of investigation and will be considered in further studies. Lastly, the weight similarities in the male and female calves of the present study are probably due to the high inter-individual variations of the female calves’ weights.

In dromedary camel species, more than 80% of the fetal membranes are expelled within two hours from delivery [24,25,26]; the observations of the present study are in line, therefore, with previous reports.

During parturition, the fetal membranes are pulled out through the umbilical cord and, after its rupture, by their own and the remaining fluids’ weight. The pregnant horn side is therefore turned inside out, whereas the part of the membranes in conjunction with the non-pregnant horn portion is pulled through its attachment with the uterine body. This pattern might explain both the features of the placenta after expulsion and the breaking points located at the base or at the middle portion of the non-pregnant horn (Appendix A).

The color of the placenta, with areas ranging from pale pink to dark red, could be due to storage-related changes: the pressure on the capillaries when placentas are placed in a plastic bag could force the blood to move from one area to another, producing patterns of dark and pale areas [14]. The dark red areas might also be due to the initial placental detachment during the delivery process. In fact, similarly to what happens in the mare (red bag), the vessels in the microcotyledonary villi might become engorged (congested) when detached before other areas of the placenta (e.g., detachment of the part corresponding to the umbilical cord attachment or at the uterine body, as reported by Schlafer [18]). Whitwell and Jeffcott [15] observed darker areas around the cord attachment and identified, microscopically, an increased distribution of the villi and their taller size. An accurate sampling and histological examination of the above-mentioned areas, soon after the placenta’s expulsion, is needed for a better understanding of these macroscopic features. These aspects will therefore be carefully considered in future studies.

Although the edema of the tip of the pregnant horn is a common finding in the mare’s placenta [3,15], in the present study it was not observed, except in one case, which was not included in the study.

Contrariwise to what was observed in mares, avillous areas such as the cervical star or areas corresponding to oviductal papilla or the folding of the chorion around the allantoic vessels were not noticed. It is noteworthy that no areas without villi were found at the medial margins of the uterine horns, where the umbilical vessels run on the allantoic surface, as was instead reported in South American camelids by Meesters et al. [16]. Sometimes, diffuse areas with more rarefied villi were observed at the tip of the uterine horns, confirming Morton’s previous observations [13]. The author reported that several areas, bare of villous tuts, could be seen mainly near the pole of the pregnant and non-pregnant horns. However, typical areolae such as the horse ones were not observed.

In another case, excluded from the present study, diffuse multiple focal round-shaped avillous areas of 1–2 cm in diameter were observed (Appendix A) without any effect on the calf’s health.

Regarding the size of chorionic surfaces, the present data indicate that the pregnant uterine horn portion is almost about one third bigger than the non-pregnant portion (159 ± 21.2 cm vs. 105.2 ± 16.83 cm, respectively). This also explains the higher correlations between the TPW and TV with the VPH. In one placenta, the pregnant horn length was found to be double that of the LNPH (175 vs. 87 cm) and five times higher in the TV as compared with the non-pregnant horn (4560 vs. 900 mL) (Figure 2(6A)). Higher values of TPW and TV were observed in male calves. However, no statistical differences were found regarding other parameters (horn linear measurements and calves BW). A higher number of observations would probably be needed to confirm differences between male and female calves in terms of body weight and placental linear measurements.

Whitwell and Jeffcott [15] observed that in 24% of the examined mare’s placenta, the non-pregnant horn size was equal or bigger than the pregnant horn, but the authors did not provide an explanation of that finding. Bravo et al. [27] stated that the caudal width of the pregnant horn portion is age-dependent in Alpaca, reaching a peak at 9 years of age and decreasing at 12. Since the feto-maternal exchanges would be improved with a higher placenta surface, it would be worth understanding the factors (age, parity, BCS) related to the size of the non-pregnant horn portion (e.g., comparing Figure 2(2A,5A)). Unfortunately, in the present study, it was not possible to retrieve the reproductive anamnesis of the recipient dams. A more detailed reproductive data collection could be performed in subsequent studies to specifically focus on these aspects.

On the allantoic side, the amniotic sac was observed mainly on the ventral part of the placenta surface but, in a few cases, also on the dorsal surface. Opposite to what was observed in the equine placenta, in which the amnion is free in the amniotic cavity [3], the amnion was found attached to the chorionallantois. Accordingly, the torsion of the umbilical cord should be considered rarer in DCs when compared with the equine species (amnion free to move in the allantoic cavity), and it could be limited only to the intraamniotic portion (i.e., the DC fetus has limited mobility due to the fixed amniotic membrane). However, there are no data regarding this aspect in the DC species. According to the findings of the present study, it is unlikely that the amnion could be delivered with the newborn, but this contrasts with previous literature [13,26].

The attachment of the umbilical cord was always found at the medial side of the left pregnant horn except in one case, excluded from the study, where it was found at the lateral side of the pregnant horn (Appendix A). This finding could indicate that different vascular patterns might be found in DCs, as observed also in the equine species [28]. A right uterine horn pregnancy cannot be excluded as being the reason for the different site of the UC attachment, but this hypothesis seems unlikely because Benaissa et al. [29] observed only one right uterine horn pregnancy out of 198 DCs’ pregnant uteri examined at the slaughterhouse.

The amniotic plaques, observed also in equine [18] and Alpaca [16] placentas, are focal areas, scattered on the fetal side of the amniotic membrane, whose surface could become keratinized and appear as small horn-like rugose growths [13,18].

The placental median weight resembles that reported by Atieh et al. [21] in DCs in Saudi Arabia’s Majaheem and Wodoh ecotypes (6.35 ± 0.43 and 6.51 ± 0.34 kg, respectively). The placenta efficiency (TPW/calve BW) was 15%. Since the ratio is higher than reported in the horse [15,30] and donkey [10,31], and considering that the dromedary camel GL is more similar to that of the donkey [32], a lower efficiency of the DC placenta might be hypothesized, as compared with the equine species.

A 10.3% efficiency of *Llama glama* placentas (dams aged between 2 to 9 years) was previously reported by Meesters et al. [16], whereas, from the study of Bravo et al. [28], the cria weight/placenta weight was around 11 as compared to 6.7 in the present study. The same author observed an increase in the FMs’ weight from 6 to 10 years of age of the dam, followed by a decline. Because of the selection of the recipients for the embryo transfer program (age between 8 to 15 years with at least one parturition) it was not possible to evaluate the relationship between their age, the weight of the obtained placentas, and the overall placenta efficiency. A higher number of specimens would be needed to assess the correlation between dam age and placental efficiency [33,34,35].

## 5. Conclusions

In conclusion, this study provides a standardized on-field evaluation of the dromedary camel normal placenta with some useful information about its gross characteristics. In order to better describe the normal and pathologic features of the dromedary placenta and to understand actual placental efficiency for the growth and development of the fetus, as reported for horses and donkeys, further investigation involving the histologic examination is required.

## Figures and Tables

**Figure 1 animals-14-01553-f001:**
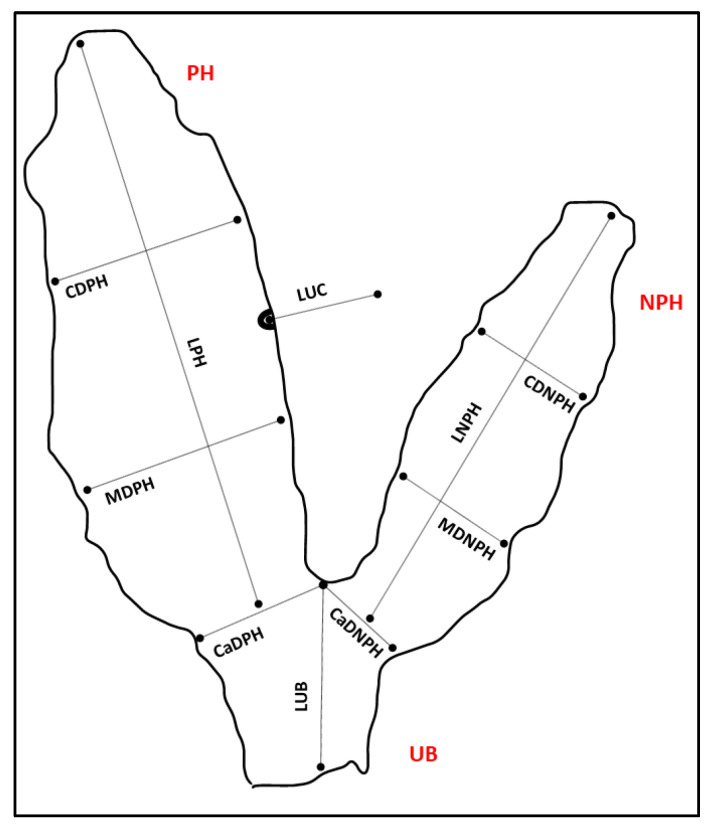
Scheme of the dromedary camel placenta positioned in a “Y” shape, for efficient examination and collection of the linear measurements. PH: pregnant horn; NPH: non-pregnant horn; UB: body of uterus; LPH: length of the pregnant horn; CDPH: cranial diameter of the pregnant horn; MDPH: middle diameter of the pregnant horn; CaDPH: caudal diameter of the pregnant horn; LNPH: length of the non-pregnant horn; CDNPH: cranial diameter of the non-pregnant horn; MDNPH: middle diameter of the non-pregnant horn; CaDNPH: caudal diameter of the non-pregnant horn; LUB: length of the body of uterus; LUC: length of the umbilical cord. For CDPH, MDPH, CaDPH, CDNPH, MDNPH, and CaDNPH, maximum diameters in the corresponding portion were considered. Following the linear measurements, the umbilical cord and the two horns were dissected, and their volumes were evaluated to the nearest 50 mL, by using a graduated bucket: (i) volume of the pregnant horn (VPH); (ii) volume of the non-pregnant horn (VNPH); (iii) volume of the uterine body (VUB); (iv) volume of the umbilical cord and vessels (VUC). The total volume (TV) of FMs was calculated afterward. The total weight of the fetal membranes (TPW) was also measured by using an electronic counting scale (model SBS-PW-301CC, expondo Polska, Zielona Góra, Poland) to the nearest 1 g, and the ratios between the TPW and the calf BW and BW/TPW were finally calculated, as a measure of placental efficiency [16].

**Figure 2 animals-14-01553-f002:**
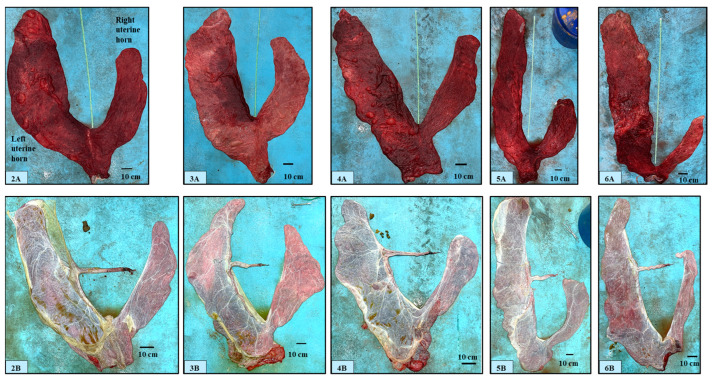
(**2**–**6**). Dromedary camel placentas: chorionic (**2A–6A**) and allantoic (**2B–6B**) surfaces.

**Figure 3 animals-14-01553-f003:**
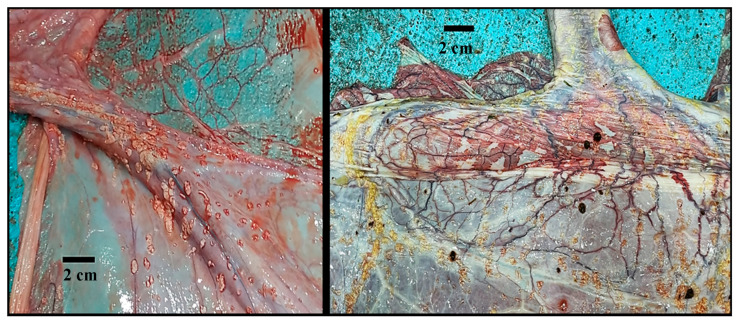
Dromedary camel placentas: amniotic plaques observed on the inner surface of the amniotic sac, close to the attachment of the umbilical cord.

**Figure 4 animals-14-01553-f004:**
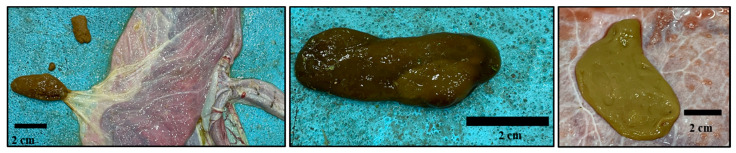
Dromedary camel placentas: camelmanes (hippomanes).

**Figure 5 animals-14-01553-f005:**
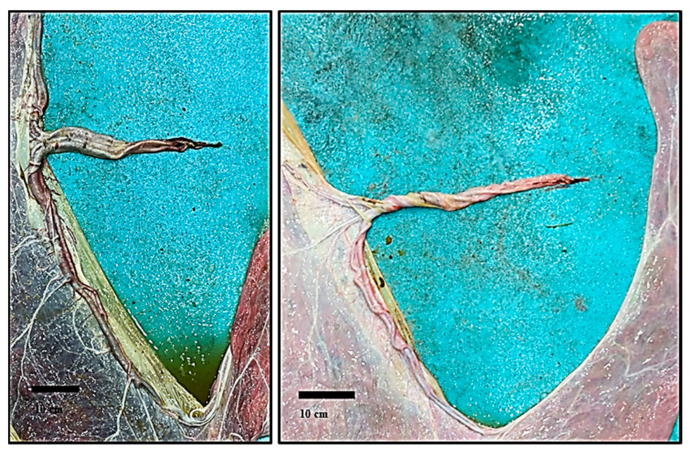
Dromedary camel placentas: attachment of the umbilical cord and umbilical vessels along the allantoic medial side of the pregnant and non-pregnant uterine horn portions.

**Figure 6 animals-14-01553-f006:**
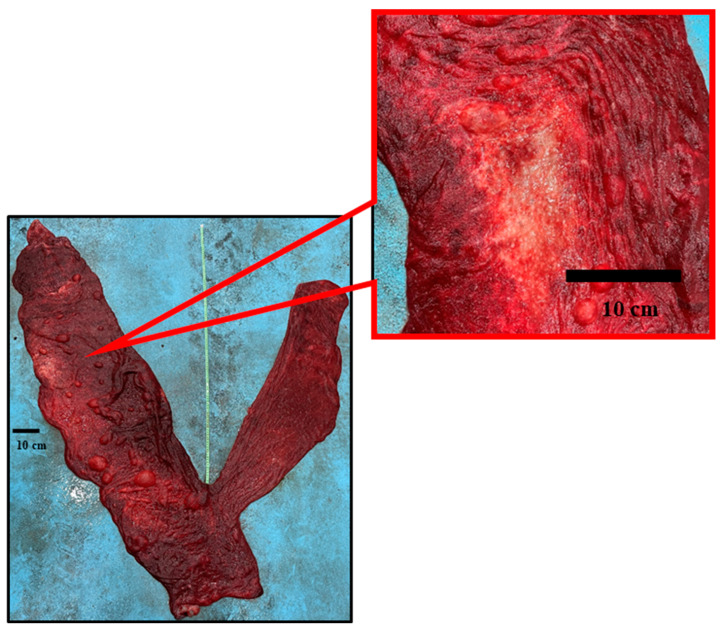
Dromedary camel placentas: avillous area and its detail (5×), attributed to a foot pad, observed on the chorionic surface.

**Table 1 animals-14-01553-t001:** Descriptive data (mean ± SD, min–max) of dromedary camel pregnancy length, calf expulsion time, placenta expulsion time, male calves BW, and female calves BW.

Pregnancy Length (Days) *n* = 11	Calf Expulsion Time (min) *n* = 11	Placenta Expulsion Time (min) *n* = 11	Male Calves BW (kg) *n* = 5	Female Calves BW (kg) *n* = 6
372.0 ± 8.9	14.83 ± 1.18	71.20 ± 36.67	44.4 ± 1.97	41.2 ± 4.38
(357–381)	(13–16.5)	(20–150)	(41.8–46.6)	(35.6–47.4)

**Table 2 animals-14-01553-t002:** Descriptive data (mean ± SD, median, min–max) of dromedary camel placentas (*n* = 11) linear measurements.

	Mean (cm) ± SD	Median (Min–Max) (cm)
LPH	159 ± 21.2	160 (119–190)
LNPH	105.2 ± 16.83	109 (81–136)
LUB	40.5 ± 6.61	41 (28–51)
CDPH	36.1 ± 5.38	35 (26–45)
MDPH	42.5 ± 6.21	41 (34–52)
CaDPH	48.8 ± 4.49	48 (42–57)
CDNPH	24.5 ± 6.22	23 (15–37)
MDNPH	25.6 ± 4.41	25 (18–33)
CaDNPH	29.8 ± 5.59	29 (21–43)
LUC	42.9 ± 7.02	44 (33–53)

LPH: length of the pregnant horn; LNPH; length of the non-pregnant horn; LUB: length of the uterine body; CDPH: cranial diameter of the pregnant horn; MDPH: middle diameter of the pregnant horn; CaDPH: caudal diameter of the pregnant horn; CDNPH: cranial diameter of the non-pregnant horn; MDNPH: middle diameter of the non-pregnant horn; CaDNPH: caudal diameter of the non-pregnant horn; LUC: length of the umbilical cord.

**Table 3 animals-14-01553-t003:** Descriptive data (mean ± SD, median, min–max) of dromedary camel placental components volumes (mL).

	Mean (mL) ±SD	Median (Min–Max) (mL)
VPH	3832.7 ± 1114.94	3500 (2450–6300)
VNPH	1475 ± 533.98	1250 (900–2500)
VUB	975 ± 313.80	1000 (600–1600)
VUC	295.5 ± 85.01	300 (200–500)
TV	6877.3 ± 1623.63	6550 (4700–10,200)

VPH: volume of the pregnant horn; VNPH: volume of the non-pregnant horn; VUB: volume of the body of uterus; VUC: volume of the umbilical cord and vessels. TV: total volume.

## Data Availability

The data supporting the findings of this study are available from the corresponding authors upon reasonable request.

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
