# Peer review of "On-field Gross Morphology Evaluation of Dromedary Camel (Camelus dromedarius) Fetal Membranes"

_animals, 2024, doi:10.3390/ani14111553_

Round 1

Reviewer 1 Report

Comments and Suggestions for Authors

See attachment file

Comments on the Quality of English Language

Basically fine, but some sentences are difficult to understand. In addition to modifying the sentence contents (see my comments), English editing may be able to solve this problem.

Author Response

Thank you so much for all suggestion

Reviewer 2 Report

Comments and Suggestions for Authors

The present work appears really interesting for veterinary practice, particularly for camelid (dromedary camel in particular) reproductive management where limited data are currently available. Generally, the work is well-conducted and the design is well-described. Following are some suggestions to improve the fluency of the text.

line 17: delete comma

line 19: delete comma

line 24: The pregnant horn FMs portion was always bigger in size than the one non-pregnant. Delete “horn”. 

line 30-31: similarly to the mare’s placenta. Delete other. 

line 31: remove “after delivery”, you specify when the evaluation of placenta must be done in the next sentence. 

line 38: delete one of the periods at the end of the sentence.

lines 45-46:  It is common, however, to refer to the foetal membranes (FMs) as “the placenta”. Consider that the foetal membranes comprise chorion, allantois, but also amnios. The latter is not a part of the placenta.

lines 49-50: and in some species, for the production and metabolism of various hormones. Remove “and in some species”. If you decide to specify in some species you should indicate which species. Please, provide another reference. The reference 1 is specific for only equine species. 

line 52: Remove the second “for”.

line 56: What do you refer to with “other”?  

line 73: Remove part in the brackets. 

line 77: Remove “after delivery”. It is already clear that in your study the FMs evaluation is done after delivery. 

line 97: Remove the two commas.

line 129: the nearest g ? Maybe a number was missed.

line 129-130. Wilson (reference n. 12) reported that the placenta efficiency may be calculated with the ratio between the offspring BW and PTW. Why did you use the contrary ratio? (PTW/BW).

Fig.2-6: Please, provide a scale bar and right/left (of the uterine horns) orientation at least in one picture.

line 160: as the line 24 comment.

Fig. 7-8. Please, provide a scale bar and add names of the different parts of placenta you evidenced in those pictures. 

Fig 12-13. as the Fig.7-8 comment.

Fig. 14.15. as the Fig. 7.8 comment.

line 224. The data about correlation is really interesting. I would like to suggest (if it possible) a correlation chart just to make it easier to follow the results with a graphic support. 

lines 231-234. Please, cut the sentence, it is too long. Maybe you can use a period before “However”. 

lines 236: Delete the last “and '' at the end of sentence. Did you calculate the ratio between BW and PTW or the contrary? Please, correct also consider  the statement at line 129-130.

line 244-245. I think it is too strong to say that it is possible to “recognize the modifications of intra-uterine metabolic exchanges” only to perform an easy on-field gross morphology evaluation. The ability of nutrient/oxygen/waste exchanges throughout the placenta may be estimated through ecodoppler analysis of blood flow at placental vessels levels, or - more specifically - by performing molecular analysis in ex-vivo tissue. Please, rewrite this sentence, avoiding linking with a possible evaluation of nutrient exchange capacity of the placenta tissue. 

line 253: maybe it is “alive”

lines 256-259. Please divide in two sentences, just to get easier the reading 

lines 270. Remove “of the foetus”

lines 298. Remove the comma before “by….”. Please, pay attention to the use of commas in the whole manuscript. 

line 299. Remove commas before and after “sometimes”, they are not necessary. 

line 298-302. Please, divide your sentence. I often notice the use of too long sentences. Please check the whole manuscript focusing on this syntactic aspect. 

Supplementary Figures. Please, provide in all pictures a scale bar and right/left (of the uterine horns) orientation. 

Author Response

Please see the attachment. Thank you for all suggestion

Reviewer 3 Report

Comments and Suggestions for Authors

The manuscript presented here reports on the morphology and size of the placenta of dromedary camels. The data presented are preliminary and require further detailed study. The reviewers were uncertain whether the data presented was novel enough information to justify publication.

Comments:

1.    Introduction: The reviewers do not know why the authors chose to focus on camels. What are the issues with placental formation, etc., in animal species, including camels? How is the camel placenta unique compared to other species and how might it contribute to the development of this field? The authors should address the description of these points.

2.    The authors should describe the significance and scientific purpose of revealing the morphology of the camel placenta. They should also state what discoveries were made as a result of the clarification.

3.    Material & Methods: The authors should provide a subtitle for the content of each paragraph.

4.    The reviewer could not find any description of how the placentas obtained were photographed. The reviewer requests a clear description.

5.    The original position criteria for the length of CDPH, MDPH, CDNPH, and MDNPH should be listed.

6.    Results: The authors should state the purpose of the experiment at the beginning.

7.    The numbering and legends of figures should be revised by referring to previous papers and instructions of journal publishers. Scale bar is missing in the figures and the size is not clear. Scale bar should be added in the figures. If it is an enlarged view, it should also be described (Fig. 14).

8.    The authors should provide a more detailed and clear and sufficient description of the placental structures obtained. For example, the authors should explain where the cervical stars and allantoic vessels are distributed and what unique characteristics they have, in linkage with the figure.

9.    Fig. 7-11: How did you know it was amnion (camelmanes?)? Are there any other possibilities? The reviewer does not know why it was identified for avillous area as well. The authors should prepare these detailed tissue specimens for observation and scientific identification. Alternatively, it should be demonstrated from a different aspect.

Comments on the Quality of English Language

Thank you for giving me an opportunity of reviewing the manuscript of animals-2983178. I am not sure that this information is sufficient to support publication in Animals. Additionally, I am uncertain whether the data presented was novel enough information to justify publication.

Author Response

(The authors gave the same response as above.)

Round 2

Reviewer 3 Report

Comments and Suggestions for Authors

8. The authors should provide a more detailed and clear and sufficient description of the placental structures obtained. For example, the authors should explain where the cervical stars and allantoic vessels are distributed and what unique characteristics they have, in linkage with the figure.

Please, could the reviewer be more specific about this comment? In the result part it is mentioned that "macroscopic avilluos areas (i.e. cervical star or in correspondence of allantoic vessels) were not noticed. This aspect was clearly considered in the discussion part with the following paragraph:

9. Fig. 7-11: How did you know it was amnion (camelmanes?)? Are there any other possibilities? The reviewer does not know why it was identified for avillous area as well. The authors should prepare these detailed tissue specimens for observation and scientific identification. Alternatively, it should be demonstrated from a different aspect.

Thank you for the note. The authors are quite sure that what is reported is camelmanes (note that the term “camelmanes” was proposed by the authors because it is the first work reported on gross macroscopic features of placenta and the dromedary camel is neither a horse nor a bovine to call it a hippomane or bovimane), both from their practical work experience and from what has been reported in the literature in the horse for example, on the hippomane, because it has peculiar characteristics. Pozor 2016 reported "There are certain characteristic features often found in the allantoic cavity called 'hippomanes' (Fig 8c) or 'allantoic calculi. In equine species contain lipids,

cellular debris, degenerated blood cells, and irregular mineralised material (Schlafer 2004)".

I am sorry that I wrote this in a confusing manner. More directly, the reviewer believes that the gross characteristics of the placenta alone are insufficient to scientifically demonstrate that it is a placenta. It is necessary to demonstrate by other scientific methods, such as physiological, micromorphological, or statistical methods that the organisms are a placenta. If, as the authors say, this is the first study reported, it should be carefully researched, and evidence should be accumulated. 

Author Response

Dear Reviewer, 

This is one of the first studies done on the evaluation of the placenta in this species. The present study was articulated as for other species in the past (horses, cattle, camelids and others). The authors think that it is normal for the first studies to examine only the macroscopic part, the part that is seen immediately after parturition. This study was designed by the authors specifically for field work and thus was intended as macroscopic only, so that it would be a quick guide for those who attend parturition on the field and thus can visually notice visible alterations. In fact, the supplementary figures include the file that can be used to examine the placenta and can serve as a guide so as not to forget any aspect, including the histological aspect, which is mentioned. The histological aspect indeed will be included in the next paper, but this first paper is the first simplest step to describe what is normal in the placenta of the dromedary camel and to put the foundation on the evaluation of the placenta in this species. Thank you!